# Survival Outcomes of Patients with Esophageal Cancer Who Did Not Proceed to Surgery after Neoadjuvant Treatment

**DOI:** 10.3390/cancers15164049

**Published:** 2023-08-10

**Authors:** Chun-Ling Chi, Xing Gao, Hsiang-Yu Hsieh, Yi-Hsuan Cheng, Zhi-Hao Yang, Yin-Kai Chao

**Affiliations:** 1Division of Thoracic Surgery, Chang Gung Memorial Hospital-Linkou, Chang Gung University, Taoyuan 333, Taiwan; irenechi3312159@icloud.com (C.-L.C.); x.gao.1@erasmusmc.n (X.G.); b0602041@gmail.com (H.-Y.H.); brian06250118@gmail.com (Y.-H.C.); chad0000408@gmail.com (Z.-H.Y.); 2Department of Surgery, Erasmus Medical Center, 3015GD Rotterdam, The Netherlands

**Keywords:** neoadjuvant treatment, esophageal cancer, esophagectomy, non-surgical treatment, patient refusal

## Abstract

**Simple Summary:**

Approximately one-third of patients with esophageal squamous cell carcinoma did not undergo surgical resection after neoadjuvant chemoradiotherapy. The overall survival of this patient group is generally less favorable, the only exception being the subgroup of those refusing surgery.

**Abstract:**

Background: This retrospective study examined outcomes in esophageal squamous cell carcinoma (ESCC) patients who did not undergo surgical resection after neoadjuvant chemoradiotherapy (nCRT). Methods: Patients receiving nCRT between 2012 and 2020 were divided into two groups: group 1 (scheduled surgery) and group 2 (no surgery). Group 2 was further categorized into subgroups based on reasons for not proceeding to surgery: group 2a (disease progression), group 2b (poor general conditions), and group 2c (patient refusal). Overall survival (OS) was the primary outcome. Results: Group 1 comprised 145 patients, while subgroups 2a, 2b, and 2c comprised 24, 16, and 31 patients, respectively. The 3-year OS rate was significantly lower in group 2 compared with group 1 (34% versus 56%, *p* < 0.001). A subgroup analysis showed varying 3-year OS rates: 13% for group 2a, 25% for group 2b, and 58% for group 2c (*p* < 0.001). Propensity score matching for group 2c and group 1 revealed no significant difference in 3-year OS rates (*p* = 0.91). Conclusion: One-third of ESCC patients receiving nCRT did not undergo surgical resection. Overall survival in this group was generally poorer, except for those who refused surgery (group 2c).

## 1. Introduction

Esophageal cancer (EC) is a highly aggressive malignancy that portends a poor prognosis [1]. Disease recurrence following primary tumor resection continues to be common despite recent surgical improvements [2]. The mainstay of treatment for locally advanced EC is neoadjuvant chemoradiotherapy (nCRT) followed by surgery within 8–12 weeks [3,4]. Significant benefits of nCRT prior to resection include tumor downstaging and the opportunity to treat upfront any undetected micrometastasis [3,5]. However, a non-negligible proportion of patients with EC who underwent nCRT do not proceed with a scheduled esophagectomy [6]. While randomized clinical trials have generally included a limited number of patients not proceeding to surgery, it is well known that they do not replicate real-world conditions of routine clinical care [6]. According to a recent analysis of population-based data, approximately 14% of patients with EC did not undergo surgical resection after starting nCRT—with this condition being associated with a decreased overall survival (OS) [7]. Unfortunately, this study did not provide insight into the reasons for not undergoing surgery.

In general, potential barriers to esophagectomy after the completion of nCRT include (1) the development of interval distant metastasis prior to resection, (2) complications of nCRT leading to poor general conditions, and (3) patient refusal in the absence of contraindications to surgery. In a publication by Depypere et al., reasons for canceling esophagectomy were disease progression in 43.9% of the cases, poor general condition in 22.8%, irresectability in 12.3%, refusal in 13.2%, and death during neoadjuvant treatment in 7.9% [8]. While the expected outcomes regarding disease progression and poor physical condition are dismal, the prognostic impact of surgery refusal has not been completely elucidated. Widespread fears of postoperative complications and impaired quality of life may lead to an unfavorable perception of esophagectomy [6,9]. Clinical improvement following nCRT may also impact esophageal surgery decision-making. Although most patients with EC who underwent nCRT can achieve a more favorable survival after esophagectomy, the adverse prognostic significance of surgery refusal remains speculative and empirically untested. This single-center study was therefore designed to investigate the survival outcomes of patients with EC not proceeding to esophagectomy after nCRT, with particular attention to the effect of the underlying reason on OS and progression-free survival (PFS).

## 2. Materials and Methods

### 2.1. Study Patients

This is a single-center retrospective analysis of prospectively collected data. IRB approval was obtained to review medical charts of patients diagnosed with non-cervical esophageal squamous cell carcinoma (ESCC) from 2012 to 2020. Patients who received nCRT as first-line treatment were eligible. Individuals with a history of other malignancies identified in the five years preceding the diagnosis of ESCC were excluded, as were subjects with multiple tumors at diagnosis. Patients were initially divided into two groups according to whether they underwent scheduled surgery (group 1) or not (group 2). Group 2 was further divided into three subgroups according to the reason for not proceeding to surgery as follows: disease progression (group 2a), poor general conditions (group 2b), and patient refusal (group 2c). A study flowchart is presented in Figure 1.

### 2.2. Indications for Neoadjuvant Chemoradiotherapy

In accordance with our institutional policy, all patients with a clinical stage of cT2-4aN0M0 or cT1-4aN + M0 were offered nCRT in the pre-operative period. Two nCRT regimens were utilized throughout the study period. The first regimen (PF regimen) consisted of 5-fluorouracil (5-FU; 1000 mg/m^2^ per day, continuously infused over 96 h from days 1 to 4 and from days 29 to 33) and cisplatin (75 mg/m^2^; administered as an intravenous infusion over 3 h on day 1 and day 29). Radiation therapy was delivered between days 8 and 29. The total dose was 41.4–45 Gy, administered in daily fractions of 180 cGy, 5 days per week [10]. Radiotherapy encompassed the entire esophagus and local regions of lymphoid tissue. The supraclavicular fossa, celiac, and pericardial lymphatic regions were also irradiated unless the delivered dose to normal tissues was not tolerable. The second regimen (CROSS regimen) was based on the weekly administration of carboplatin (doses titrated to achieve an area under the curve of 2 mg per milliliter per minute) and paclitaxel (50 mg/m^2^ body surface area) for 5 weeks and concurrent radiotherapy (41.4–45 Gy in 23–25 fractions, 5 days per week). Radiation was delivered using paired anterior and posterior treatment portals or with intensity-modulated radiotherapy.

### 2.3. Prestaging and Restaging Workup

Pretreatment staging and restaging evaluation were based on computed tomography (CT) of the chest and abdomen, esophagography, endoscopic ultrasound (EUS), and positron emission tomography (PET). EUS was performed with an ultrasonic miniprobe (UM2R/12 MHz or UM3R/20 MHz; Olympus, Tokyo, Japan). Patients were staged according to the 2010 (7th) AJCC staging criteria. Patient performance status was assessed with the Eastern Cooperative Oncology Group (ECOG) scale, ranging from 0 (fully active, able to carry out all pre-disease performance without restriction) to 5 (dead) [11]. The severity of comorbidities was calculated with the Charlson comorbidity index (CCI), a measure of the overall disease burden based on the presence of 19 distinct medical disease categories. Each comorbidity is weighted (from 1 to 6) according to the degree to which it predicts mortality. The total score ranges from 0 to 37, with a score of zero indicating the absence of comorbidities.

A clinical restaging workup was scheduled at 4–6 weeks after completion of nCRT. Post-treatment evaluations were based on the results of physical examination, endoscopic biopsies, and imaging investigations—including thoracic and abdominal CT and PET scans.

### 2.4. Assessment of Tumor Response

The assessment of tumor response was carried out by a multidisciplinary team using the Response Evaluation Criteria in Solid Tumors (RECIST) for CT images and the Positron Emission Tomography Response Criteria in Solid Tumors (PERCIST) criteria for PET images [12]. According to RECIST, objective response to treatment is divided into four categories: complete response (CR), partial response (PR), progressive disease (PD), and stable disease (SD). The four PERCIST categories of progressive metabolic disease (PMD), stable metabolic disease (SMD), partial metabolic response (PMR), and complete metabolic response (CMR) were also assigned [13]. The following criteria were used to define a complete tumor response: (1) absence of malignant cells in biopsy specimens, (2) achievement of CR or PR on CT images, and (3) achievement of CMR on PET/CT scans. A complete lymph node response was defined by the simultaneous presence of CR (lesion shrinkage to <10 mm) on CT images and CMR (resolution of fluorodeoxyglucose uptake to a level indistinguishable from surrounding normal tissues) on PET/CT scans. When a primary tumor or nodal residual disease was undetectable based on the results of both endoscopic and radiological imaging investigations, a clinical complete response (cCR) was considered to be present.

### 2.5. Surgery and Additional Treatment Modalities

In the absence of contraindications, esophagectomy was scheduled at 6–8 weeks after completion of nCRT. Patients were considered operable if they were physically fit for surgery and had no evidence of either tracheoesophageal fistula or recurrent laryngeal nerve invasion. The standard surgical approach applied throughout the study period consisted of a transthoracic esophagectomy with intrathoracic gastric tube reconstruction (Ivor Lewis procedure) or cervical anastomosis (McKeown procedure). While all operated patients underwent two-field lymph node dissection, cervical lymphadenectomy was selectively performed in those who had evidence of residual disease in the cervical area. Patients who refused surgery were encouraged to undergo consolidation therapy. The final decision was made after discussion and agreement with a multidisciplinary team. Most patients who were unwilling to undergo surgery were treated with an additional course of CRT—for which the same chemotherapy regimen used for nCRT was given. As for RT, the additional dose ranged between 23.4 and 30 Gy. All patients were regularly followed up every 3–4 months for the first two years, every 6 months between the third and the fifth year, and on a yearly basis thereafter [14].

### 2.6. Statistical Analysis

All continuous data were expressed as means and standard deviations (SDs), whereas categorical variables were given as counts and percentages. Continuous variable comparisons were performed with the Student’s *t*-test (normally distributed data) or the Mann–Whitney U test (skewed data). Categorical variables were assessed using the χ^2^ test or Fisher’s exact test when the expected cell count was less than 5. Propensity score (PS) matching was implemented to minimize any imbalance between group 1 and subgroup 2c with respect to age, sex, body mass index, clinical stage (according to the UICC Cancer Staging Manual, eighth edition) [15], tumor length, tumor location, Charlson’s comorbidity index, chemotherapy regimen, weight loss during nCRT, clinical tumor response, clinical lymph node response, and achievement of cCR. Based on the nearest estimated value on the logit score, two PS-matched cohorts with a 1:3 ratio were created using the optimal pairing method [16]. Standardized mean differences (SMDs) were calculated to detect any remaining imbalance in PS-matched groups, with values < 0.10 being considered as indicative of a good balance. OS and PFS were defined as the interval from the end of nCRT to the day of death or progression of the disease, respectively. The progression date was determined retrospectively by examining patients’ charts, specifically when signs of recurrence or progression were observed on CT scans or endoscopy. Analyses were conducted using the ‘tableone’, ‘MatchIt’, and ‘stats’ packages implemented in R, version 3.6.1 (R Foundation for Statistical Computing, Vienna, Austria). All hypothesis testing was two-sided, and statistical significance was set at *p* < 0.05.

## 3. Results

### 3.1. General Characteristics of the Study Patients

The study cohort consisted of 216 patients with ESCC (mean age: 57 years). Of them, 145 underwent scheduled surgery (group 1), whereas 71 did not (group 2). No significant intergroup difference was observed in terms of general and nCRT-related characteristics (Table 1), although patients in group 2 tended to be older (56.28 ± 8.68 versus 58.55 ± 8.45 years, respectively; *p* = 0.073) and more commonly received the PF regimen (*p* = 0.097).

### 3.2. Survival Outcomes

The median follow-up time for the entire cohort was 40.18 months. The 3-year OS rate was significantly lower in group 2 than in group 1 (34% versus 56%, respectively; *p* < 0.001). In addition, the median survival time (MST) was significantly lower in the former group (17 months) compared with the latter (54 months; *p* < 0.001; Figure 2). On analyzing group 2 according to the reasons for not proceeding to esophagectomy after nCRT, the following three subgroups were identified: disease progression (group 2a; *n* = 24), poor general conditions (group 2b; *n* = 16), and patient refusal (group 2c; *n* = 31). The 3-year OS rates for the three subgroups were 13%, 25%, and 58%, respectively (*p* < 0.001; Figure 3).

### 3.3. Survival Outcomes of Group 1 versus Subgroup 2c after Propensity Score Matching

After the implementation of PS matching, group 1 and subgroup 2c showed a good balance in terms of clinical characteristics, burden of comorbidities, and clinical stage (Table 2). In the PS-matched cohort, the 3-year OS rates of patients who underwent esophagectomy and those who refused surgery were 59% and 61%, respectively (*p* = 0.91; Figure 4a). Similar findings were observed in terms of 3-year PFS rates (45% versus 42%, respectively, *p* = 0.56; Figure 4b). Patients who refused surgery showed a higher disease progression rate (58%) than those who received scheduled esophagectomy (43%). As for the location of disease progression, three patients (10%) in subgroup 2c progressed at the primary tumor site only; conversely, 21 patients (22%) in group 1 had evidence of distant metastasis only (Table 3). Of the six patients who had locoregional disease progression alone, four were offered salvage surgery, but R0 resection was feasible in one case only.

## 4. Discussion

In this single-center retrospective cohort study, we found that a considerable proportion (33%, 76/228) of patients with ESCC did not ultimately proceed to scheduled surgery after completion of nCRT. When patients who did not undergo esophagectomy were analyzed as a whole, their 3-year OS rates were less favorable than those who did—an observation consistent with prior findings [8]. However, once we assessed whether the reasons for not proceeding to surgery translated into different survival outcomes, we observed that the 3-year OS rate of patients who voluntarily refused esophagectomy (irrespective of the presence of contraindications) was similar to that of patients who received the scheduled resection. Importantly, this observation was confirmed and extended to PFS after PS matching to minimize any imbalance between group 1 and subgroup 2c with respect to clinical and pathological characteristics. In the era of shared decision-making, our data provide important prognostic information that can impact thoracic surgery evaluations when patients with ESCC who underwent nCRT express their unwillingness to proceed with esophagectomy—due to their own values, perceptions, and outcome expectations.

In a previous population-based study, Rahouma et al. [17]. showed evidence of less favorable survival outcomes in patients with EC who refused surgery. However, their conclusions relied on patients who declined to undergo esophagectomy when they were still treatment-naïve. In light of baseline intergroup differences with respect to factors associated with refusing surgery—especially in terms of vulnerability—the results of Rahouma et al. [17] are not surprising. Instead, we focused on patients with ESCC who initially accepted surgery as part of their multimodal treatment but modified their own decision and declined to undergo esophagectomy only after completion of nCRT. Second, a detailed chart review allowed us to undertake a separate analysis of patients who did not proceed to surgery due to poor general conditions. This methodological approach would not have been feasible with a population-based design such as that used by Rahouma et al. [17].

An organ-sparing curative approach for EC remains an actively debated topic, with the focus of research predominantly targeting two methods [18]. The first method involves the selection of patients with a cCR for active surveillance, while the second method advocates for the enhancement of radiation dosage to a potentially curative magnitude in all patients, bypassing the need for surgery. The challenge of the former method lies in the selection of appropriate patients, emphasizing the criticality for accurate clinical response assessment. On the contrary, the latter method requires the optimization of local disease control. Research findings reveal that only 23–40% of cCR patients possess no residual cancer upon pathological examination and thus do not accurately reflect the presence of a pCR [19,20,21]. Nevertheless, publications have suggested that the survival outcomes of organ-sparing strategies can be non-inferior to those achieved with scheduled esophagectomy after nCRT [6,22,23]. Unfortunately, definitive guidelines for active surveillance are not yet available [24].

Concerning the second method, previous studies conducted in the context of definitive or consolidation treatment have indicated that high-dose chemoradiotherapy could potentially yield survival outcomes comparable to those of nCRT followed by surgery [25,26]. More precisely, a randomized trial by Stahl et al. in 2005 inferred that definitive chemoradiotherapy (at least 65 Gy) and chemoradiotherapy followed by surgery (40 Gy) yield similar OS rates [27]. Adding surgery to chemoradiotherapy improves local tumor control but does not increase survival of patients with locally advanced ESCC. The same rationale was used in the FFCD 9102 trial, where consolidation therapy (an additional 15–20 Gy) supplemented nCRT as an alternative to surgery, demonstrating no benefit of surgical intervention compared to the continuation of additional chemoradiotherapy [28].

In the current study, not all patients who declined surgery achieved a cCR, and the application of an active surveillance protocol consisting of bite-on-bite endoscopic biopsies was not invariably feasible. Nonetheless, respectable survival rates were observed in those who refused surgery compared to those who underwent resection. At least two explanations can account for this finding. First, misclassification of patients who were actually free from residual cancer (i.e., cCR and pCR) into the non-cCR category might have occurred as a result of the limited accuracy of imaging investigations [29]. Second, the vast majority (90%) of patients who voluntarily refused surgery did not merely undergo surveillance but was treated with consolidation therapy [30]. Even though not all patients in subgroup 2c achieved cCR, they generally responded well to nCRT and, as a result, were anticipated to exhibit a favorable response to consolidation therapy as well.

While our cohort of patients who willingly declined surgery, as well as the non-surgery groups in the aforementioned trials, demonstrated commendable survival rates, the potential for further improvement remains a viable consideration. One considerable limitation of this approach relates to the harsh outcomes of salvage surgery, typically the subsequent step in managing locoregional recurrence (LR). Of the six patients in subgroup 2c who had evidence of LR alone, four were offered salvage surgery, but R0 resection was feasible in one case only. These results call for a thorough follow-up surveillance aimed at timely detection of relapsing disease when scheduled surgery following nCRT is voluntarily refused or, as previously discussed, the suitable selection of patients devoid of residual disease. Simultaneously, it is vital to contemplate the counter hypothesis that surgery does not augment mortality due to postoperative complications and hence, remains the recommended therapeutic intervention following nCRT until further empirical evidence suggests otherwise.

Some limitations of our data should be considered. The number of patients who refused surgery against the physicians’ recommendations was relatively limited, and replication of our findings in other studies is necessary. However, the reasons for refusal are not noted in the patients’ charts and could therefore not be discussed. While we know from our results that patients who made the decision to decline surgery after nCRT can achieve respectable survival outcomes, we should acknowledge that shared guidelines or standard approaches for their clinical management are still missing. Developing a consensus on how to deliver consolidation therapy to this patient group is clearly important to minimize prognostic uncertainty.

## 5. Conclusions

Approximately one-third of patients with ESCC who underwent nCRT do not ultimately proceed to surgical resection. The OS of this patient group is generally less favorable, the only exception being the subgroup of those refusing surgery.

## Figures and Tables

**Figure 1 cancers-15-04049-f001:**
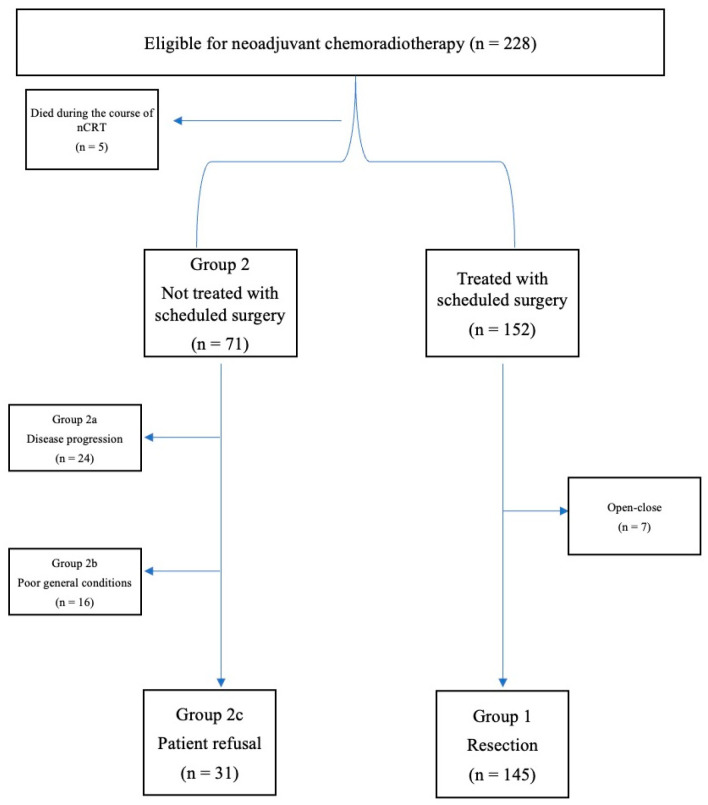
Flow of patients through the study.

**Figure 2 cancers-15-04049-f002:**
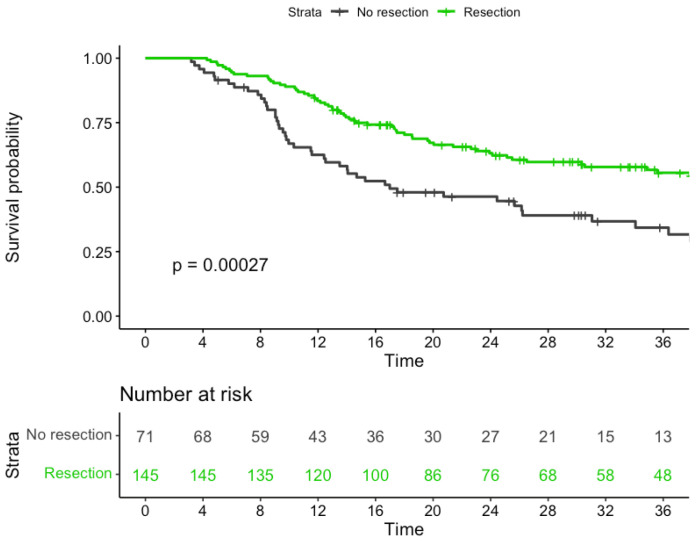
Kaplan-Meier curves of overall survival in patients who received resection or not.

**Figure 3 cancers-15-04049-f003:**
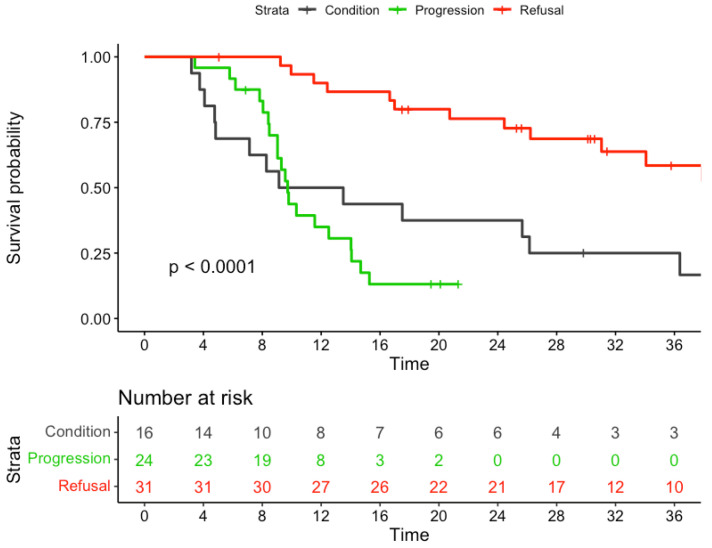
Kaplan–Meier curves for overall survival of patients in group 2 stratified by their reason for no resection.

**Figure 4 cancers-15-04049-f004:**
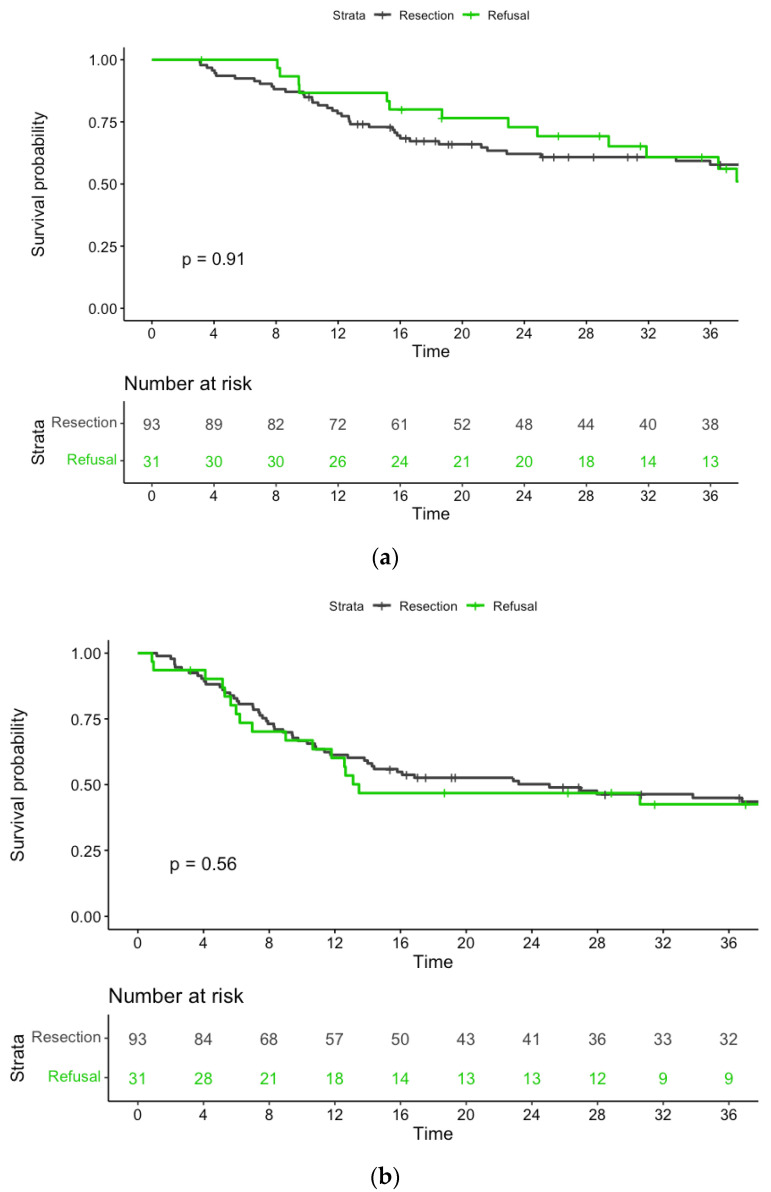
(**a**) Kaplan–Meier curves for overall survival after PSM. (**b**) Kaplan–Meier curves of progression-free survival after PSM.

**Table 1 cancers-15-04049-t001:** Baseline characteristics of the study patients.

		Group 1	Group 2	*p*
		*n* = 145	*n* = 71	
**Baseline characteristics**				
Age, years		56.28 (8.68)	58.55 (8.45)	0.073
Men, *n* (%)		139 (95.9)	67 (94.4)	0.883
BMI, kg/m^2^		22.71 (3.61)	22.34 (3.36)	0.466
Clinical stage				0.896
	II, *n* (%)	18 (12.4)	9 (12.7)	
	III, *n* (%)	103 (71.0)	52 (73.2)	
	IVa, *n* (%)	24 (16.6)	10 (14.1)	
Tumor length, cm		5.66 (2.62)	5.73 (2.57)	0.847
Tumor location				0.201
	Lower third, *n* (%)	53 (36.6)	22 (31.0)	
	Middle third, *n* (%)	66 (45.5)	41 (57.7)	
	Upper third, *n* (%)	26 (17.9)	8 (11.3)	
CCI		2.55 (1.24)	2.65 (1.14)	0.582
**Neoadjuvant treatment**				
Chemotherapy				0.097
	CROSS, *n* (%)	111 (76.6)	46 (64.8)	
	PF, *n* (%)	34 (23.4)	25 (35.2)	
Radiotherapy dose, Gy		43.89 (3.21)	44.42 (1.42)	0.191
Duration of nCRT, days		48.93 (11.57)	51.86 (18.79)	0.160
Weight loss during nCRT				0.236
	<5%, *n* (%)	103 (71.0)	57 (80.3)	
	5–10%, *n* (%)	27 (18.6)	7 (9.9)	
	>10%, *n* (%)	15 (10.3)	7 (9.9)	

Patients were divided into two groups according to whether they underwent scheduled surgery (group 1) or not (group 2). Data are presented as means (standard deviations) unless otherwise indicated. Abbreviations: BMI, body mass index; CCI, Charlson’s comorbidity index; CROSS, carboplatin/paclitaxel; PF, 5-fluorouracil/cisplatin; nCRT, neoadjuvant chemoradiotherapy.

**Table 2 cancers-15-04049-t002:** Propensity score-matched cohort of patients who underwent scheduled surgery (group 1) or did not because of patient refusal (group 2c).

		Group 1 (before PSM)	Group 1 (after PSM)	Group 2 (Refusal)	*p* (after PSM)	SMD (after PSM)
		*n* = 145	*n* = 93	*n* = 31		
**Baseline characteristics**						
Age, years		56.28 (8.68)	58.70 (8.82)	58.76 (7.14)	0.970	0.009
Men, *n* (%)		139 (95.9)	87 (93.5)	29 (93.5)	1.000	0.000
BMI, kg/m^2^		22.71 (3.61)	22.87 (3.50)	22.82 (2.62)	0.946	−0.018
Clinical stage					0.801	
	II, *n* (%)	18 (12.4)	11 (11.8)	5 (16.1)		0.117
	III, *n* (%)	103 (71.0)	71 (76.3)	23 (74.2)		−0.049
	IVa, *n* (%)	24 (16.6)	11 (11.8)	3 (9.7)		−0.073
Tumor length, cm		5.66 (2.62)	5.94 (2.56)	5.94 (2.51)	0.995	−0.001
Tumor location					0.920	
	Lower third, *n* (%)	53 (36.6)	33 (35.5)	12 (38.7)		0.066
	Middle third, *n* (%)	66 (45.5)	49 (52.7)	16 (51.6)		−0.022
	Upper third, *n* (%)	26 (17.9)	11 (11.8)	3 (9.7)		−0.073
CCI		2.55 (1.24)	2.71 (1.26)	2.71 (0.90)	1.000	0.000
**Neoadjuvant treatment**						
Chemotherapy					0.381	
	CROSS, *n* (%)	111 (76.6)	64 (68.8)	18 (58.1)		−0.218
	PF, *n* (%)	34 (23.4)	29 (31.2)	13 (41.9)		0.218
Radiotherapy dose, Gy		43.89 (3.21)	43.85 (3.4)	44.25 (2.4)	0.559	0.160
Duration of nCRT, days		48.93 (11.57)	49.65 (10.49)	48.68 (11.86)	0.668	−0.081
Weight loss during nCRT					0.979	
	<5%, *n* (%)	103 (71.0)	67 (72.0)	23 (74.2)		0.049
	5–10%, *n* (%)	27 (18.6)	18 (19.4)	5 (16.1)		−0.09
	>10%, *n* (%)	15 (10.3)	8 (8.6)	3 (9.7)		0.036
**Response evaluation**						
* cCR (tumor)		28(19.3)	18 (19.4)	7 (22.6)	0.897	0.077
* cCR (lymph nodes)		38(26.2)	18 (19.4)	6 (19.4)	1.000	0.000
* cCR (ycT0N0M0)		34(23.4)	4 (4.3)	2 (6.5)	1.000	0.088

* Included in propensity score matching. Abbreviations: PSM = propensity score matching; SD = standard deviation; BMI = body mass index; CCI = Charlson’s comorbidity index; nCRT = neoadjuvant chemoradiotherapy; CROSS = carboplatin/paclitaxel; PF = 5-fluorouracil/cisplatin; cCR = clinical complete response.

**Table 3 cancers-15-04049-t003:** Disease progression after refusing surgery.

	Refusal	Resection (after PSM)	*p* (after PSM)
	*n* = 31	*n* = 93	
**Additional therapy**			
Chemoradiotherapy (%)	28 (90)	-	*-*
Time from end of nCRT to additional treatment (median days [IQR])	53 (43.5–63.5)		
Surveillance (%)	3 (10)	-	*-*
**Salvage surgery (%)**	4 (13)	-	*-*
**R0 resection (%)**	-	88 (95)	*-*
**Disease progression (%)**	18 (58)	40 (43)	0.146
**Disease progression within 1 y (%)**	10 (32)	28 (30)	0.822
**Location of disease progression**			0.017
No progression	13 (42)	53 (57)	
Locoregional metastasis	14 (45)	19 (21)	
without distant metastasis	6 (19)	13 (14)	
with distant metastasis	8 (26)	6 (7)	
Distant metastasis only	4 (13)	21 (22)	

Data are presented as counts (percentage). Abbreviations: PSM = propensity score matching; nCRT = neoadjuvant chemoradiotherapy; IQR = interquartile range.

## Data Availability

Data can be provided upon request.

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
