# Peer review of "Survival Outcomes of Patients with Esophageal Cancer Who Did Not Proceed to Surgery after Neoadjuvant Treatment"

_cancers, 2023, doi:10.3390/cancers15164049_

Round 1

Reviewer 1 Report

Thank you for this paper. I have some comments/questions,

1) Do you have data on OS and PFS stratified by who had pathologic complete response vs. partial response vs. no response on the surgical group and could then compare to Group 2? woudl be interesting to see what your response rates were in the surgical group.

2) Do you have a group 2c who refused surgery that Did NOT receive any additional chemo or XRT? that woudl be important, b/c many do not undergo additional chemo or XRT after the initial CROSS trial dose due to toxicity concerns? how many group 2c patients got addtional treatment vs. those that did not  and how did that impact survival and PFS?

3) were the authors able to look at charts specifically for PFS? how was that obtained? 

Thank you.

Author Response

The figure is in the Word file.

1) Do you have data on OS and PFS stratified by who had pathologic complete response vs. partial response vs. no response on the surgical group and could then compare to Group 2? Would be interesting to see what your response rates were in the surgical group

Thank you for the interesting input. We attempted to stratify patients in the surgery group based on their response evaluation. However, only two patients were scored as having no response, as the progressed patients were already separated. As a result, almost all patients showed some response on the primary tumor site, if not in the lymph nodes. When we stratified the patients based solely on the response of the primary tumor site, the survival rates remained the same across the four groups (see figure).

Nevertheless, we considered these findings to be clinically irrelevant because the response of the lymph nodes plays a significant role in a patient's survival. We apologize for not being able to include this information in our manuscript.

2) Do you have a group 2c who refused surgery that Did NOT receive any additional chemo or XRT ? that would be important , b / c many do not undergo additional chemo or XRT after the initial CROSS trial dose due to toxicity concerns ? how many group 2c patients got additional treatment vs. those that did not and how did that impact survival and PFS ?

Thank you for addressing the concern. As mentioned in Table 3, out of 31 patients, only 3 did not receive any additional treatment. Unfortunately, due to the limited sample size, conducting a survival analysis was not feasible. However, we were able to gather data on the average time it took for these patients to start the additional treatment after the completion of neoadjuvant chemoradiotherapy (nCRT). We have included this information in Table 3 of the revised manuscript, showing a median of 53 days with an interquartile range of 43.5-63.5 days.

3) Were the authors able to look at charts specifically for PFS ? how was that obtained ?

Yes, charts were reviewed for each patient to obtain the progression-free survival (PFS) data. In the revised manuscript, we have provide additional details on how we acquired the PFS data: “The progression date was determined retrospectively by examining patients' charts, specifically when signs of recurrence or progression were observed on CT scans or endoscopy.”

Reviewer 2 Report

In a retrospective study, the authors analyzed the OS and PFS of the patients who did not receive the esophagectomy after neoadjuvant chemoradiotherapy (NACRT). The authors concluded that the OS of this patient group is generally less favorable except for the subgroup of those refusing surgery. There are some reports about patients without surgery after neoadjuvant CRT. Most reports mentioned that the NACRT with surgery group is better or no more than the NACRT without surgery. This report was similar to these reports. Furthermore, the study design has some improvements. Although the main aim was an interesting clinical question, there needed to be some corrections to make this conclusion and for publication. 

Major Comments

1, The Kaplan-Meier analysis needed to have the censored points. The survival rate was inconsistent. Although 46 patients out of 146 survived, the OS of the resection group was above 50% at 36 months. No information regarding censoring has been provided. The authors should mention the median-follow period and handle the censored appropriately.

2, The details of background factors need to be clarified in subgroups. Mainly the authors analyzed OS and PFS with propensity score matching (PSM) between Group1 and Group2c. The authors should mention their background factors before PSM. There was no significant difference between group 1 and group 2. Group 2c included the patients with a good condition in PS or tumor factors.

3, What reasons for refusal are there in the refusal group? In this particular study, it is crucial to understand the reasons why some patients did not undergo surgery. Without this information, there is a risk of unfairly undermining the validity of the decision to refuse surgery. Therefore, it is essential to consider and analyze the factors contributing to the choice of non-surgical treatment options in order to provide a comprehensive and balanced assessment of the study results. Although the CR rate was 22.6% in the refusal group, it was guessed that most patients had a good response to NACRT. The authors should add the details of the reasons for refusal.

Minor Comments

1, In line 70, the authors defined the indication of NACRT as T2N0M0 or higher. This indication was unclear, for example, the patients of T1bN1or2M0. The authors add some information at this point.

2, The authors indicated in the discussion that Six patients received salvage surgery after local progression in the refusal group. This information is an essential factor for OS. The authors should mention this in the result rather than the discussion.

3. In the abstract, the study period was from 2012 to 2018. In material and method, that was 2012-2020. Which is correct?

Author Response

1 )The Kaplan - Meier analysis needed to have the censored points . The survival rate was inconsistent. Although 46 patients out of 146 survived, the OS of the resection group was above 50 % at 36 months . No information regarding censoring has been provided. The authors should mention the median - follow period and handle the censored appropriately.

We apologize for not providing information on censored points, you are absolutely correct about this importance. In the revised version, we have included information on censored data and replaced the Kaplan-Meier curves with KM-curves that indicates censoring. The median follow-up time, calculated from the reverse KM curve, is 40.18 months and has been added to the manuscript.

2) The details of background factors need to be clarified in A subgroups. Mainly the authors analyzed OS and PFS with propensity score matching (PSM) between Group1 and Group2c. The authors should mention their background factors before PSM. There was no significant difference between group 1 and group 2. Group 2c included the patients with a good condition in PS or tumor factors.

Thank you for your suggestion regarding clarifying the background factors before propensity score matching (PSM). If you meant the baseline characteristics, we have included the baseline characteristics of group 1 in Table 2 for better comparison. However, it is worth noting that these characteristics were already mentioned in Table 1, except for the response characteristics.

3) What reasons for refusal are there in the refusal group? In this particular study , it is crucial to understand the reasons why some patients did not undergo surgery . Without this information, there is a risk of unfairly undermining the validity of the decision to refuse surgery. Therefore, it is essential to consider and analyze the factors contributing to the choice of non - surgical treatment options in order to provide a comprehensive and balanced assessment of the study results . Although the CR rate was 22.6 % in the refusal group, it was guessed that most patients had a good response to NACRT . The authors should add the details of the reasons for refusal.

We agree that providing insight into the reasons behind patients' decisions is crucial for a balanced assessment of the study results. However, in our patient's report, those reasons are often not mentioned, except of course for cases where the general condition is weak or if there's disease progression. We have added this information to our limitations: "The reasons for refusal are not noted in the patients’ charts and could therefore not be discussed."

Minor Comments

1) In line 70, the authors defined the indication of NACRT as T2NOMO or higher . This indication was unclear, for example, the patients of T1bN1or2M0 . The authors add some information at this point.

You are correct, we have changed this into: all patients with a clinical stage of cT2-4aN0M0 or cT1-4aN+M0 were offered nCRT.

2 ) The authors indicated in the discussion that six patients received salvage surgery after local progression in the Refusal group . This information is an essential factor for OS. The authors should mention this in the result rather than the discussion.

Thank you for this reminder. According to our manuscript, 4 patients received salvage surgery. We have added this to our results section as well

3) In the abstract, the study period was from 2012 to 2018 . material and method , that was 2012-2020 . Which is correct?

We apologize for this error. The study period from 2012-2020 is correct. We have amended this in the revised manuscript.

Reviewer 3 Report

The authors bring a well written manuscript evaluating outcomes of patient with M0 disease not undergoing resection after preoperative chemotherapy and irradiation for SCC of the esophagus..  The authors conclude that resection should be uniformly considered after preoperative therapy in this setting.

I have a few questions/queries for the authors:

1. How was response determined for cases which did not undergo resection?

2. Comparing patients with progression of disease during preoperative therapy to those without is not appropriate unless the authors recommend routine resection regardless of metastatic status. This should be clarified.

3. Comparing patients with medical contraindications to surgery to those without is not appropriate unless the authors recommend routine resection regardless of metastatic status. This should be clarified.

4. The outcomes for patients refusing surgery was essentially the same as for those undergoing resection. The contrary hypothesis suggesting that surgery does not have a major impact on selected patients must be considered and discussed.

5. There are randomized controlled German trials showing no effect of resection after preoperative therapy. These should be part of the discussion.

adequate

Author Response

1. How was response determined for cases which did not undergo resection?

In the current study, we exclusively focused on patients who had the initial treatment plan of nCRT followed by surgery. All of them received response evaluations in the same way. Per our institutional protocol and as described in our section” assessment of tumor response”. In short, with endoscopy with biopsies, CT scan and a PET scan.

2. Comparing patients with medical contraindications to surgery to those without is not appropriate unless the authors recommend routine resection regardless of metastatic status. This should be clarified.

Thanks for your comment. In the current study, we focused on locally advanced patients who were initially medical fit and with the initial treatment plan of nCRT followed by surgery. However, some of them did not receive surgery due to medical condition deterioration after nCRT or disease progression or refusal.

Whether the reason of non-surgery impact survival remained unknown.

This single-center study was therefore designed to investigate the survival outcomes of patients with EC not proceeding to esophagectomy after nCRT, with particular attention to the effect of the underlying reason on OS and progression-free survival. Sorry for the misleading.

3. The outcomes for patients refusing surgery were essentially same as for those undergoing resection. The contrary hypothesis suggesting that surgery does not have a major impact on selected patients must be considered and discussed.

Thank you for this insightful comment. The discussion section has been amended and this hypothesis is mentioned in the revised version of the discussion: “Simultaneously, it is vital to contemplate the counter-hypothesis that surgery does not augment mortality due to postoperative complications, and hence, remains the recommended therapeutic intervention following nCRT until further empirical evidence suggests otherwise.”

4. There are randomized controlled German trials showing no effect of resection after preoperatively therapy. These should be into discussion.

Thank you for your suggestion. We have amended our discussion section with a more comprehensive discussion on existing literature

Round 2

Reviewer 1 Report

Thank you for your responses